# GROUNDED COMPOSITIONAL GENERALIZATION WITH ENVIRONMENT INTERACTIONS

## ABSTRACT

In this paper, we present a compositional generalization approach in grounded agent instruction learning. Compositional generalization is an important part of human intelligence, but current neural network models do not have such ability. This is more complicated in multi-modal problems with grounding. Our proposed approach has two main ideas. First, we use interactions between agent and the environment to find components in the output. Second, we apply entropy regularization to learn corresponding input components for each output component. The results show the proposed approach significantly outperforms baselines in most tasks, with more than 25% absolute average accuracy increase. We also investigate the impact of entropy regularization and other changes with ablation study. We hope this work is the first step to address grounded compositional generalization, and it will be helpful in advancing artificial intelligence research. The source code is included in supplementary material.

## 1 INTRODUCTION

Compositional generalization is a key skill for flexible and efficient learning. Humans leverage compositionality to create and recognize new combinations of familiar concepts (Chomsky, 1957; Minsky, 1986). Though there are many progresses for machine learning and deep learning in various areas recently (LeCun et al., 2015), current main learning algorithms are not able to perform compositional generalization, and require many samples to train models. Such efficient learning is even more important when machines interact with the environment for grounding, because interactions are usually slow.

Machine learning has been mostly developed with an assumption that training and test distributions are identical. Compositional generalization, however, is a kind of out-of-distribution generalization (Bengio, 2017), where training and test distributions are different. During training, dataset does not contain the information of the difference, so it can only be given as prior knowledge. In compositional generalization, a sample is a combination of several components. Test distribution changes as test samples are new combinations of seen components in training. For example, if we can find "large apple" and "small orange" in some environments, then we can also find "large orange" among multiple objects in a new environment.

The recombination is enabled when an output component depends only on the corresponding input components, and invariant of other components (please see Section 4.1 for more details). So there are two aspects to consider. What are the components in output, and how to find the corresponding input signals. We propose to use interactions between agent and the environment to define output components. This is analogues to model-free reinforcement learning (Sutton & Barto, 2018), where an agent does not have an environment model, but leans to act at each step during the interactions with the environment. Then we use entropy regularization (Li et al., 2019; Li & Eisner, 2019) to learn the minimal input components for outputs.

We evaluate the approach with gSCAN dataset (Ruis et al., 2020), which is designed to study compositional generalization in grounded agent instruction learning. Please see Figure 1 for examples. The results show the proposed approach significantly outperforms baselines in most tasks, with more than 25% absolute average accuracy increase, and the high accuracy indicates that the proposed approach addresses the designed grounded compositional generalization problems in these tasks. We also look into the impact of entropy regularization and other changes with ablation study.

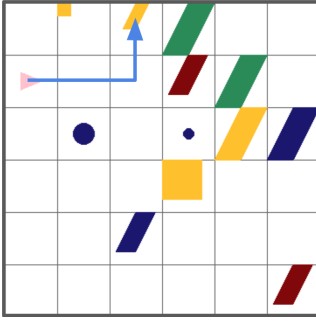 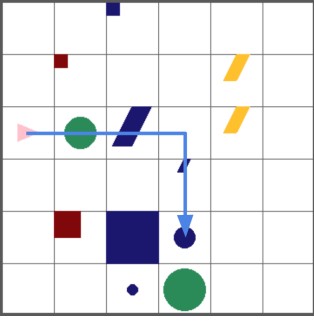

(a) Walk to a small cylinder.  (b) Walk to a large blue circle.

Figure 1: Examples of gSCAN dataset. The dataset evaluates compositional generalization ability in grounded instruction learning. The agent needs to understand command and the environment to take sequence of actions. Training and test data have different distributions and tasks require different kinds of compositional generalization. Please refer to Section 5 for more details.

We hope this work will be helpful in advancing grounded compositional generalization and artificial intelligence research.

The contributions of this paper can be summarized as follows.

- This is the first work to enable accurate compositional generalization in grounded instruction learning problem, serving for analyzing and understanding the mechanism.
- The novelty of this paper is to find that the combination of environment interaction and entropy regularization helps the generalization.

## 2 RELATED WORK

Compositional generalization research has a long history, and recently there are increasing focus on this area. SCAN dataset (Lake, 2019) was proposed to study compositional generalization in instruction learning. It maps a command sentence to a sequence of actions. This dataset has a property that input words and output actions have direct correspondence. Though some NLP tasks, such as machine translation, have similar property, not all problems fit to the setting. Also this dataset does not contain an environment for an agent to take actions.

SCAN dataset inspired multiple approaches (Russin et al., 2019; Andreas, 2019; Li et al., 2019; Lake, 2019; Gordon et al., 2020; Liu et al., 2020). Some of them lead to general techniques for compositional generalization. For example, entropy regularization (Li et al., 2019) is proposed to avoid redundant dependency on input, and it is a core idea of the approach in this paper.

Compositional generalization has applications in various fields such as question answering (Andreas et al., 2016; Hudson & Manning, 2019; Keysers et al., 2020), counting (Rodriguez & Wiles, 1998; Weiss et al., 2018), systematic behaviour (Wong & Wang, 2007; Brakel & Frank, 2009), and hierarchical structure (Linzen et al., 2016). Another related work is independent disentangled representation (Higgins et al., 2017; Locatello et al., 2019), but they do not address compositional generalization. Compositionality is also helpful for reasoning (Talmor et al., 2020) and continual learning (Jin et al., 2020; Li et al., 2020).

Grounded SCAN (gSCAN) dataset was proposed to introduce environment and grounding to agent instruction learning with compositional generalization (Ruis et al., 2020). It has a command sentence as input and a sequence of actions as output. However, the input command does not tell the specific way to act, but agent needs to understand the environment and take corresponding actions. This also avoids direct mapping between input words and output actions.

Different approaches have been proposed to address this problem. As compositional generalization requires prior knowledge for distribution change, these approaches correspond to different ways to provide the prior knowledge. Andreas (2019) uses linguistic knowledge to augment training data.

Kuo et al. (2020) uses external syntactic parser and WordNet. In this paper, we apply the prior knowledge for the interactions of agent and the environment.

# 3  PROBLEM DESCRIPTION

gSCAN dataset contains episodes of commands and actions, and each episode has an input and an output. The input includes a command and an environment. A command is a sequence of words. An environment contains an agent and a set of objects. An agent has its position and initial direction. An object has its position and attributes of color, shape and size. The output contains a sequence of actions (Figure 1).

We hope to extract prior knowledge from environment interactions. We first notice that an agent should know whether it is going to change position. This means we can break an episode to a sequence of steps, where each step corresponds to an action for position change. An agent should also know the change of directions between steps. So we separate the direction change and the manner of action in a step. Therefore, we have three output for each step: direction, action and manner. We further convert the environment to be agent centered, and rotate the environment to make agent facing forward. We also assume automatic collision prevention. If the agent tries to push or pull an object to collide with other objects or boarder, then it stops. This makes us focus on addressing grounded compositional generalization problem.

In summary, when we consider agent interaction with the environment, we can convert the problem to a set of step-wise label prediction problems with multiple outputs. Input contains command, environment, state, and output contains direction, action and manner.

# 4  APPROACH

In this section, we describe the approach for grounded compositional generalization. As we use entropy regularization in different modules, we first introduce it, then move to the model architecture.

## 4.1  ENTROPY REGULARIZATION

The difficulty of compositional generalization is that there might be incorrect dependency between input and output components. For example, when "red" and "square" do not appear together in any training sample, a model might learn that square is not red. However, this causes errors for compositional generalization in test. To avoid such case, we hope the representation of shape not influenced by input information of color.

Entropy regularization (Li et al., 2019; Li & Eisner, 2019) aims at reducing entropy of a representation to avoid dependency on redundant input components. Given an representation $x$, we compute the $L_2$ norm and add normal noise to each element of the representation. This decreases the channel capacity, so that the entropy for the representation reduces. We then feed the noised representation to the next layer, and add the norm to loss function.

$$\mathcal{L} = \mathcal{L}_{\text{original}} + \lambda L_2(x) \qquad\qquad \text{EntReg}(x) = x + \alpha\mathcal{N}(0, I)$$

where $\alpha$ is a weight of noise, positive for training and zero for inference.

A representation can be fed to multiple networks, requiring different regularization for each input node. So we design entropy regularization layer, where we achieve non-linear mapping by expanding each node $x_i$ to a vector $h_i \in \mathbb{R}^H$ with ReLU activation, and maps it back to a node $y_i$ with linear activation, then apply entropy regularization on $y$ to get $y'$. The input $x$ and output $\text{ERL}(x) = y'$ have the same size.

$$h_i = \text{ff}_i^A(x_i, H), \qquad\qquad y_i = \text{ff}_i^B(h_i, 1), \qquad\qquad y' = \text{EntReg}(y).$$

We write $\text{ff}(x, K)$ for feed-forward neural network with $x$ as input and $K$ as output size. We use EntReg for word embeddings and ERL for environment inputs.

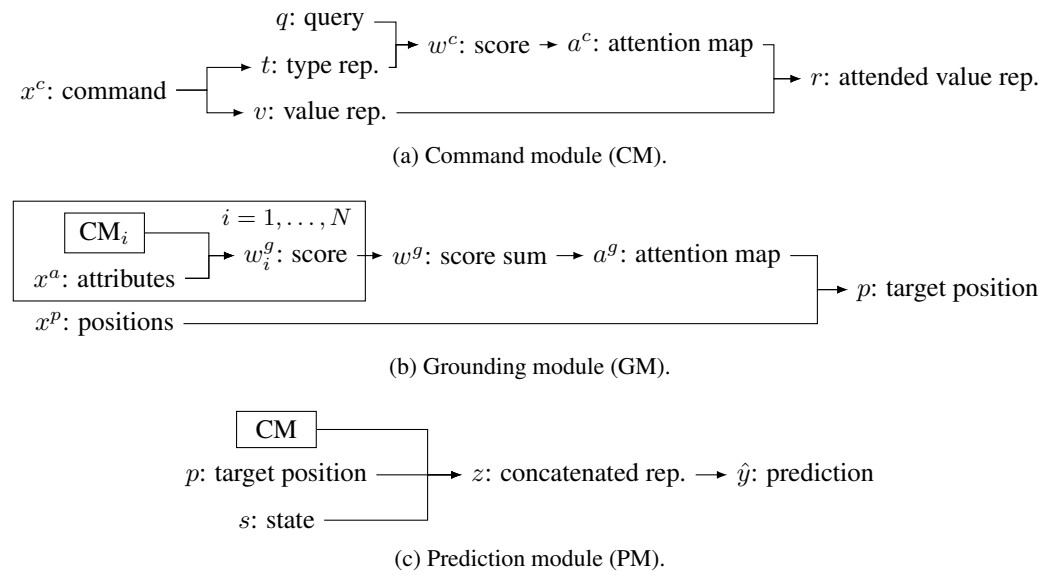

(a) Command module (CM).

(b) Grounding module (GM).

(c) Prediction module (PM).

Figure 2: Flow charts for core part of each module in the proposed approach. Command module and grounding module are based on attention mechanism as the main architecture, and prediction module concatenates inputs and use feed-forward network. More details can be found in Algorithm 1.

## 4.2 MODEL ARCHITECTURE

Input contains one-hot representation of a command $x^c$ with sentence length $n$ and vocabulary size $V^c$, and a sequence of $m$ objects. We denote object attributes $x^a$ with vector size $V^a$, and object positions $x^p$ with vector size $V^p$. We also has a binary state $s \in \{0, 1\}$ indicating whether it is the first step of an episode. Output has three types. Direction $y^d$ has $C^d$ classes, action $y^a$ has $C^a$ classes, and manner $y^m$ has $C^m$ classes.

$$x^c = x_1^c, \ldots, x_n^c \in \mathbb{R}^{V^c \times n}, \quad x^a = x_1^a, \ldots, x_m^a \in \mathbb{R}^{V^a \times m}, \quad x^p = x_1^p, \ldots, x_m^p \in \mathbb{R}^{V^p \times m}.$$

The model has the following modules, as summarized in Figure 2 and Algorithm 1.

**Command module (CM)** This module takes a command $x^c$ as input and returns a representation $r$ with size $K$, $r = \text{CM}(x^c, K)$. $r$ is expected to be the embedding of certain type of keyword, e.g. action, color, etc. To enable compositional generalization, we hope to separate representations of types (e.g. color) and values (e.g. red). We use types for attention maps and values for attended values in attention mechanism. For a new combination of values, the types are still recognizable, so that attention maps are correct and the corresponding values are extracted (Li et al., 2019). We design that each word $x_i^c$ in command $x^c$ has two embeddings for type $t_i$ and value $v_i$ with corresponding embedding matrices, $E_t \in \mathbb{R}^{k_t \times V^c}, E_v \in \mathbb{R}^{k_v \times V^c}$, where $k_t$ and $k_v$ are embedding sizes, respectively. We then apply entropy regularization on both $t$ and $v$ to reduce redundant dependency.

$$t = \text{EntReg}(E_t x^c) \in \mathbb{R}^{k_t \times n}, \qquad v = \text{EntReg}(E_v x^c) \in \mathbb{R}^{k_v \times n}.$$

We use attention mechanism with a query $q \in \mathbb{R}^{k_t}$ as learnable parameters, keys $t$, values $v$, and temperature $\tau^c \in \mathbb{R}$. We compute score $w^c$ and attention map $a^c$. The attended value $u$ is fed to a feed-forward network with output size $K$.

$$w^c = qt \in \mathbb{R}^n, \quad a^c = \text{Softmax}(w^c/\tau^c) \in \mathbb{R}^n, \quad u = va^c \in \mathbb{R}^{k_v}, \quad r = \text{ff}_\theta(u, K) \in \mathbb{R}^K.$$

**Grounding module (GM)** This module finds target in the environment according to the command. We use $N$ command modules for queries. For each query $r_i$, we have an dedicated entropy regularization layer on attributes $x_i'^a$, because different queries correspond to different type of attributes, and other queries would be redundant. We then compute a score $w_i^g$. For $i = 1, \ldots, N$,

$$r_i = \text{CM}(x^c, V^a) \in \mathbb{R}^{V^a} \qquad x_i'^a = \text{ERL}(x^a) \in \mathbb{R}^{V^a \times m}, \qquad w_i^g = r_i x_i'^a \in \mathbb{R}^m.$$

---

**Algorithm 1** The proposed approach. Input includes command $x^c$, state $s$, object attributes $x^a$ and object positions $x^p$. $K$ is an embedding size. Please see Section 4 for more information.

---

| Command module | Grounding module | Prediction module |
|---|---|---|
| **Input:** $x^c, K$ | **Input:** $x^c, x^a, x^p$ | **Input:** $x^c, x^a, x^p, s, K$ |
| **Output:** $r$: embedding | **Output:** $p$: target position | **Output:** $\hat{y}$: prediction |
| 1: $t = \text{EntReg}(E_t x^c)$ | 1: **for** $i = 1, \ldots, N$ **do** | 1: $p = \text{GM}(x^c, x^a, x^p)$ |
| 2: $v = \text{EntReg}(E_v x^c)$ | 2: $\quad r_i = \text{CM}(x^c, V^a)$ | 2: **for** each output type $i$ **do** |
| 3: $w^c = qt$ | 3: $\quad x_i'^a = \text{ERL}(x^a)$ | 3: $\quad k_i = \text{CM}(x^c, K)$ |
| 4: $a^c = \text{Softmax}(w^c/\tau^c)$ | 4: $\quad w_i^g = r_i x_i'^a$ | 4: $\quad$ **for** each node $j$ **do** |
| 5: $u = v a^c$ | 5: **end for** | 5: $\quad\quad p_{i,j}' = \text{ERL}(p)$ |
| 6: $r = \text{ff}_\theta(u, K)$ | 6: $w^g = \sum_{i=1}^N w_i^g$ | 6: $\quad\quad z_{i,j} = [k_i^T, p_{i,j}'^T, s^T]^T$ |
| | 7: $a^g = \text{Softmax}(w^g/\tau^g)$ | 7: $\quad\quad l_{i,j} = \text{ff}_{i,j}^\phi(z_{i,j}, 1)$ |
| | 8: $p = x^p a^g$ | 8: $\quad$ **end for** |
| | | 9: $\quad \hat{y}_i = \text{Softmax}(l_i)$ |
| | | 10: **end for** |

---

The scores are added as $w^g$. Attention map $a^g$ is computed by Softmax with temperature $\tau^g \in \mathbb{R}$, and is applied to get the attended object position $p$.

$$w^g = \sum_{i=1}^N w_i^g \in \mathbb{R}^m, \qquad a^g = \text{Softmax}(w^g/\tau^g) \in \mathbb{R}^m, \qquad p = x^p a^g \in \mathbb{R}^{V^p}.$$

**Prediction module (PM)**  Prediction module takes command, environment and state as input, and outputs a prediction. We have three separate prediction modules for direction, action and manner, respectively. Modules correspond to different keywords but share the same grounded target and state. So for each prediction module, we use one command module to extract a keyword $k$ with size $K$ for the prediction. We also have a environment module for target object position $p$, and all the prediction modules share it as an input. There is a input of state $s$, and this is also a shared input for each prediction module.

We build a dedicated feed-forward neural network from input to each output node without weight sharing. In each network of output type $i$ and node $j$, we use entropy regularization layer for target position $p$. This is because different output nodes may need to be computed from different input components, and other components are redundant, so we hope to reduce dependency of each output node to input nodes (more discussion in Section 6.1). Then all the inputs are concatenated to form a vector $z$ with size $L = K + V^p + 1$.

$$k_i = \text{CM}(x^c, K) \in \mathbb{R}^K, \qquad p_{i,j}' = \text{ERL}(p) \in \mathbb{R}^{V^p}, \qquad z_{i,j} = [k_i^T, p_{i,j}'^T, s^T]^T \in \mathbb{R}^L.$$

We then feed it to another feed-forward network to get a output node $l_{i,j}$. They are concatenated to form a logit $l_i$, and we use Softmax to output $\hat{y}_i$. $C_i$ is the number of classes for the output type.

$$l_{i,j} = \text{ff}_{i,j}^\phi(z_{i,j}, 1) \in \mathbb{R}, \qquad l_i = [l_{i,1}, \ldots, l_{i,C_i}] \in \mathbb{R}^{C_i}, \qquad \hat{y}_i = \text{Softmax}(l_i) \in \mathbb{R}^{C_i}$$

We use cross entropy and the norms for entropy regularization with weight $\lambda$ as training objective.

## 5  EXPERIMENTS

We use all eight tasks in gSCAN for experiment. Figure 1 shows examples of two gSCAN commands in different environments. We run experiments to show that the designed compositional generalization problems are correctly addressed. The tasks are summarized as follows.

A: *Random* is from the same distribution as training data. It does not contain compositional generalization, and is used to compare with other tasks as reference.

B: *Yellow squares* has a new way of referring to a type of object, e.g., "small square" in training, but "yellow square" only in test.

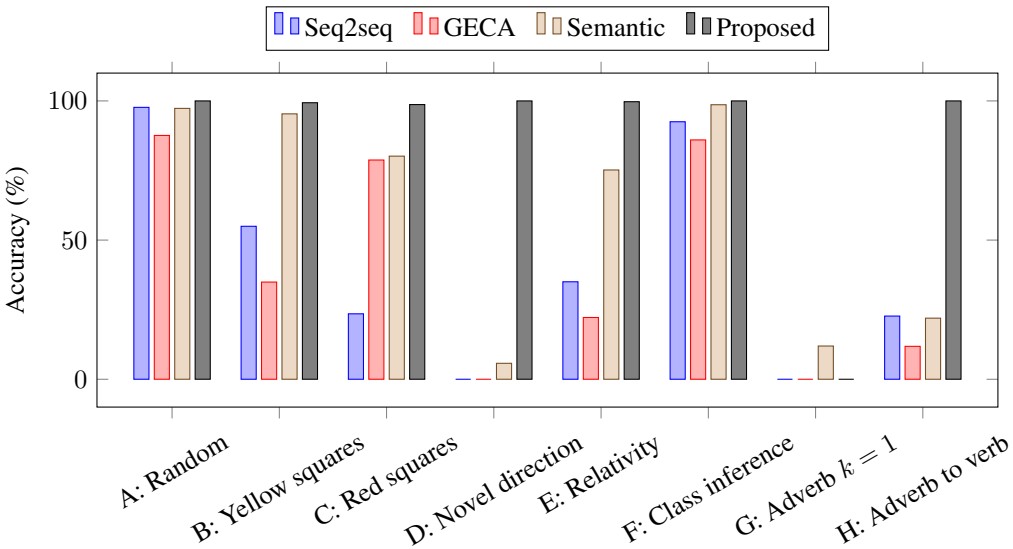

Figure 3: Bar graph of accuracy results. Seq2seq and GECA are from Ruis et al. (2020). Semantic is from Kuo et al. (2020). The results shows that the proposed approach can effectively address most of the tasks, and task G is analyzed in discussion section.

C: *Red squares* has a new combination of attribute and object, e.g., "red" and "square" appears separately in training, but "red square" appears only in test.

D: *Novel direction* has target object in a new direction, e.g., down left direction only in test.

E: *Relativity* newly has objects of specific sizes as "small" or "large". e.g., refer size 2 as "large" (a sample does not contain larger sizes) only in test.

F: *Class inference* requires inferring object properties, e.g., the object size decides the number of pull or push actions to take.

G: *Adverbs* requires learning adverbs, e.g., "cautiously", with one or a few samples.

H: *Adverb to verb* has new combination of adverb and verb, e.g., "walking" and "while spinning" both appear in training, but "walking while spinning" appears only in test.

The dataset statistics and more details of settings in model design and training can be found in Appendix A. We evaluate accuracy on episodes. An episode is correct when all the steps are correct. Following Ruis et al. (2020), we run each experiment three times, and report mean and standard deviation (note that their sum may be above 100%).

The results are shown in Figure 3 and more details in Table 1. The proposed approach has significantly higher accuracy than the baseline methods in most of the tasks. It does not work in Experiment G, and we discuss about it in Section 6.3. This indicates that the approach can effectively address most of the designed compositional generalization problems. Especially in Experiment D: Novel direction and Experiment H: Adverb to verb, there are more than 94% and 78% absolute improvements, respectively. Experiment D requires recombining different directions, and experiment H requires recombining verbs and adverbs. These tasks are standard compositional generalization problems with recombination of seen components.

# 6 DISCUSSIONS

In this section, we discuss and analyze more details to have better understanding of the approach. We will first run ablation study to find what factors enable the improvement. We then analyze the grounding ability of the proposed approach. We also discuss the one-shot learning problem. We further apply the approach on target length task.

Table 1: Details of accuracy results (%) for experiments. We report mean and standard deviations.

|  | Seq2seq | GECA | Semantic | Proposed |
|---|---|---|---|---|
| A: Random | $97.69 \pm 0.22$ | $87.60 \pm 1.19$ | 97.32 | $100.00 \pm 0.00$ |
| B: Yellow squares | $54.96 \pm 39.39$ | $34.92 \pm 39.30$ | 95.35 | $99.35 \pm 0.84$ |
| C: Red squares | $23.51 \pm 21.82$ | $78.77 \pm 6.63$ | 80.16 | $98.69 \pm 1.85$ |
| D: Novel direction | $0.00 \pm 0.00$ | $0.00 \pm 0.00$ | 5.73 | $100.00 \pm 0.00$ |
| E: Relativity | $35.02 \pm 2.35$ | $33.19 \pm 3.69$ | 75.19 | $99.72 \pm 0.38$ |
| F: Class inference | $92.52 \pm 6.75$ | $85.99 \pm 0.85$ | 98.63 | $100.00 \pm 0.00$ |
| G: Adverb $k = 1$ | $0.00 \pm 0.00$ | $0.00 \pm 0.00$ | 11.94 | $0.00 \pm 0.00$ |
| H: Adverb to verb | $22.70 \pm 4.59$ | $11.83 \pm 0.31$ | 21.95 | $100.00 \pm 0.00$ |

## 6.1 ABLATION STUDY

We conduct ablation study to understand what factors of the algorithm are important. One change of the approach is entropy regularization (EntReg). We also have structure change to apply the regularization (ERL). We have element-wise prediction for each output node (NodeOut). We design experiments by removing each of these factors.

The results in Table 2 show that the ablation experiments have lower accuracy than the proposed approach. This means all the changes are important for the approach. The reduction is especially significant for Experiment D: Novel direction. In this task, each value of the output depends on different input components, because different output values have priorities in the setting. When target appears on back, back right or back left, the agent turns around. When it appears on right, the agent turns right. Setting agent faces horizontal positive direction, this means turning around node depends only on one coordinate (horizontal is negative), but turn right node depends on two coordinates (horizontal is zero and vertical is negative). This is different from previous compositional generalization where all values of an output component depend on the same input component. Though this particular case may be solved in other ways, it is general that different output values may depend on different input components, and the proposed method is able to address the problem.

Table 2: Accuracy results (%) for ablation study. The ablation experiments have lower accuracy than the proposed approach. This means all the changes are important for the approach.

|  | Proposed | No EntReg | No ERL | No NodeOut |
|---|---|---|---|---|
| A: Random | $100.00 \pm 0.00$ | $100.00 \pm 0.00$ | $100.00 \pm 0.00$ | $95.51 \pm 6.35$ |
| B: Yellow squares | $99.35 \pm 0.84$ | $94.60 \pm 4.37$ | $74.50 \pm 0.57$ | $95.73 \pm 5.20$ |
| C: Red squares | $98.69 \pm 1.85$ | $98.27 \pm 1.38$ | $94.72 \pm 2.96$ | $96.40 \pm 5.09$ |
| D: Novel direction | $100.00 \pm 0.00$ | $92.44 \pm 7.03$ | $13.29 \pm 14.63$ | $0.00 \pm 0.00$ |
| E: Relativity | $99.72 \pm 0.38$ | $99.69 \pm 0.43$ | $96.57 \pm 3.17$ | $95.98 \pm 5.67$ |
| F: Class inference | $100.00 \pm 0.00$ | $100.00 \pm 0.00$ | $99.99 \pm 0.01$ | $95.76 \pm 6.00$ |
| G: Adverb $k = 1$ | $0.00 \pm 0.00$ | $0.00 \pm 0.00$ | $0.00 \pm 0.00$ | $0.00 \pm 0.00$ |
| H: Adverb to verb | $100.00 \pm 0.00$ | $100.00 \pm 0.00$ | $100.00 \pm 0.00$ | $96.67 \pm 4.71$ |

## 6.2 GROUNDING

We run visualization to show that the approach works in the expected way. The grounding finds correct object according to descriptions in command. A common way for grounding, which is also used in the proposed approach, is to use attention. So we evaluate whether the correct target object is attended.

The results in Table 3 show that the proposed approach has good grounding ability in all the tasks. Note that it also works on Experiment G, where the final prediction is not accurate. This means the errors in Experiment G are not caused by grounding, and the proposed approach addresses the designed grounding problems in all the tasks of gSCAN.

Table 3: Grounding accuracy results (%) for the proposed method. Grounding works well in all the tasks.

| | | | | |
|---|---|---|---|---|
| A: Random | $100.00 \pm 0.00$ | | E: Relativity | $100.00 \pm 0.00$ |
| B: Yellow squares | $100.00 \pm 0.00$ | | F: Class inference | $100.00 \pm 0.00$ |
| C: Red squares | $98.91 \pm 1.54$ | | G: Adverb $k = 1$ | $98.68 \pm 0.88$ |
| D: Novel direction | $100.00 \pm 0.00$ | | H: Adverb to verb | $100.00 \pm 0.00$ |

### 6.3 ONE-SHOT LEARNING

The gSCAN experiment result shows that Experiment G: Adverb does not work well for methods without external lexical information. This experiment was designed as few-shot or one-shot ($k = 1$) learning task, however it is different from standard settings. Conventionally, one-shot learning has only one sample for a class, but the frequency or weight of the sample is not fixed. For example, in Lake & Baroni (2018), the one-shot sample is repeated to be 10% of the whole dataset. Changing the weight of one-shot samples is also common in human learning, as we focus on one special sample and learn from it. However, in experiment G, the frequency of the one-shot sample is one, while there are more 300,000 other training samples.

This not only extremely lower the contribution of the sample to prediction loss, but also decrease the chance to be included in a mini-batch for training. To some extent, this requires the algorithm to partially address catastrophic forgetting (Kirkpatrick et al., 2017). This is because when the last mini-batches do not contain the sample, the model should not forget it. So this experiment contains more challenges than core compositional generalization, and they are beyond the scope of this paper.

### 6.4 TARGET LENGTH TASK

Ruis et al. (2020) also contains a separate dataset for target length task. It requires the model to perform on the target action sequence longer than those in training. This is not the focus of this paper, but we run experiments for references.

The result in Table 4 shows that, for the proposed approach, the accuracy does not drastically decrease when the evaluation target length increases. This might be because of using step-wise prediction, which is less influenced by the length of an episode. The result indicates an additional benefit of using environment interactions.

Table 4: Accuracy results (%) for target length experiments. Seq2seq is from Ruis et al. (2020). Semantic is from Kuo et al. (2020). The proposed approach avoids drastic decrease when the length increases.

| Target length | Seq2seq | Semantic | Proposed |
|---|---|---|---|
| 15 | $94.98 \pm 0.12$ | 93.43 | $94.45 \pm 4.56$ |
| 16 | $19.32 \pm 0.02$ | 90.88 | $96.28 \pm 3.36$ |
| 17 | $1.71 \pm 0.38$ | 87.92 | $96.04 \pm 3.25$ |
| $\geq 18$ | $< 1.00$ | 56.50 | $97.64 \pm 2.06$ |

## 7 CONCLUSION

We propose an approach to address compositional generalization in grounded agent instruction learning. We use interactions between agent and the environment to define output components, and entropy regularization to reduce redundant dependency on input. We achieve significant improvements in most of gSCAN tasks, and the high accuracy indicates that the proposed approach addresses the designed grounded compositional generalization problems in these tasks. In ablation study, we show the effectiveness of entropy regularization and other changes, and look into different aspects of the approach. We hope this work will be a step towards addressing compositional generalization in grounded language learning and general artificial intelligence.

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

## A EXPERIMENT SETTINGS

We implement the model as follows. We use one layer feed-forward neural network for $\text{ff}^A$ and $\text{ff}^B$ with $H = 4$. $\text{ff}^A$ has ReLU activation and $\text{ff}^B$ has linear activation. In command module, we use $k_t = 32$ and $k_v = 32$. $\text{ff}_\theta$ is a one layer feed-forward network with linear activation. $\tau^c = 1$. In grounding module, we use $N = 3$ and $\tau^g = 0.1$. In prediction module, we use $K = 8$. $\text{ff}^\phi_{\cdot,\cdot}$ is a three layer feed-forward network. The first and second layers have 32 hidden nodes with Relu activation. The third layer has one output node with linear activation. For target length task, the setting is the same except $N = 8$ in grounding module.

In training, we use $\lambda = 1$ and $\alpha = 0.1$. We use Adam optimizer (Kingma & Ba, 2014) with 10,000 training steps. Each mini-batch has 256 samples selected from training data uniformly at random with replacement. Gradient is clip by norm of 1. We use initial learning rate 0.001, which exponentially decays every 100 steps by a factor of 0.96. TensorFlow (Abadi et al., 2016) is used for implementation.

The sample size of gSCAN dataset is summarized in Table 5.

Table 5: Sample size of gSCAN dataset.

| Task | Samples |
|---|---|
| Train | 367,933 |
| A: Random | 19,282 |
| B: Yellow squares | 18,718 |
| C: Red squares | 37,436 |
| D: Novel direction | 88,642 |
| E: Relativity | 16,808 |
| F: Class inference | 11,460 |
| G: Adverb $k = 1$ | 112,880 |
| H: Adverb to verb | 38,582 |

## B FEW-SHOT LEARNING

The results of other few-shot learning settings in Experiment G are summarized in Table 6. More discussions for the task can be found in Section 6.3.

Table 6: Accuracy results (%) for few-shot learning experiments. Seq2seq is from Ruis et al. (2020). Semantic is from Kuo et al. (2020).

| | Seq2seq | Semantic | Proposed |
|---|---|---|---|
| Adverb $k = 1$ | $0.00 \pm 0.00$ | 11.94 | $0.00 \pm 0.00$ |
| Adverb $k = 5$ | $0.47 \pm 0.14$ | 10.17 | $0.00 \pm 0.00$ |
| Adverb $k = 10$ | $2.04 \pm 0.95$ | 33.28 | $0.00 \pm 0.00$ |
| Adverb $k = 15$ | - | 40.78 | $0.00 \pm 0.00$ |
| Adverb $k = 50$ | $4.63 \pm 2.08$ | - | $67.44 \pm 28.27$ |

