# OpenReview forum: "Grounded Compositional Generalization with Environment Interactions"
_ICLR.cc/2021/Conference — Reject_

### Official Review · AnonReviewer1 · 2020-10-27
**the paper needs major clarity improvements; reject**

**Rating:** 3
**Confidence:** 3

**Review:**

The paper aims to improve compositional generalization of grounded language learning methods. As far as I understood, this is achieved by (a) crafting a task-specific architecture with noise addition in certain spots and (b) changing the output format. Unfortunately, I did not understand many details that would be crucial to properly review this paper. I found the paper to be very unclearly written, see details below.

1. As far as I understand the key contribution of the paper is supposed to be using “interactions between the agent and the environment to find components in the output”. The place in the paper that seems to explain what this is about is Section 3. But since this section does not explain what the original action space in the task is and does not show example trajectories, I found it impossible to understand what the "components in the output" and "interactions between the agent and the environment" are.
2. The entropy regularization is also explained poorly. I understand what EntReg is. But to explain ERL the paper refers to “nodes x_i”. I do not think the paper clearly explains what these “nodes” are, and how this is related to the possibility that “a representation can be fed to multiple networks”.
3. I struggled to understand the specific architecture that is explained in Algorithm 1, but one thing that struck me is that unlike the baselines that the paper compares to, it uses an object-centric representation, whereas the baselines are taking images (albeit symbolic) as inputs. This distinction alone in principle might explain the difference in the results.
4. To make Algorithm 1 more understandable it could be helpful to visualize what the N command modules are and what they are supposed to do. A figure with an example that explains the motivation for the architecture would be of great help.

Unfortunately, this concludes my review: I was not able to understand enough in order to make more substantial comments. I would encourage the authors to improve the paper in the following aspects:
- clear explanation of the task, including input and output format
- clear presentation of the model, including a visual motivating example
- clear comparison to the baselines that takes into account the difference in input representations.

---

> ### Author Response · Authors · 2020-11-20
> **Reply to Reviewer 1**
>
> Thank you for your feedback.
>
> Q1: As far as I understand the key contribution of the paper is supposed to be using “interactions between the agent and the environment to find components in the output”. The place in the paper that seems to explain what this is about is Section 3. But since this section does not explain what the original action space in the task is and does not show example trajectories, I found it impossible to understand what the "components in the output" and "interactions between the agent and the environment" are.
>
> A1: The main contributions are stated at the end of introduction: serving for analyzing and understanding the mechanism, and finding environment interaction and entropy regularization helps. Some information of the task refers to the dataset paper, and we like to make the setting more clear in update.
>
> Q2: The entropy regularization is also explained poorly. I understand what EntReg is. But to explain ERL the paper refers to “nodes x_i”. I do not think the paper clearly explains what these “nodes” are, and how this is related to the possibility that “a representation can be fed to multiple networks”.
>
> A2: A node is a scalar element in a representation. A representation can be used as input to different networks with different outputs, and we can use ERL on the input of each network, so that each network has specific input information for its output.
>
> Q3: I struggled to understand the specific architecture that is explained in Algorithm 1, but one thing that struck me is that unlike the baselines that the paper compares to, it uses an object-centric representation, whereas the baselines are taking images (albeit symbolic) as inputs. This distinction alone in principle might explain the difference in the results.
>
> A3: The architectures are common ones. CM ad GM uses attention mechanisms. We design that the object-centric conversion is a part of the environment interaction. Also, the main focus of the paper is not comparing with other methods, but to analyze and understand the basic mechanism for the generalization.
>
> Q4: To make Algorithm 1 more understandable it could be helpful to visualize what the N command modules are and what they are supposed to do. A figure with an example that explains the motivation for the architecture would be of great help.
>
> A4: A command module produces a word from command as input to other modules. We would investigate visualization and figure for motivation.

---

### Official Review · AnonReviewer2 · 2020-10-28

**Rating:** 5
**Confidence:** 4

**Review:**

The paper proposes a new regularization method that constrains the mapping between the inputs and output spaces for achieving compositional generalization in simple grounded environments like gSCAN. The problem is interesting and important and the paper is corroborated by good experiments with 25% accuracy increase and also generalization to longer commands. However, the paper has  clarity issues with descriptions that are sometimes vague or not precise enough and quite frequent language mistakes. It also doesn't discuss almost at all existing works and mention the only very briefly, making it harder to judge the strength of the new approach as there's not enough context.

In addition, the idea presented feels somewhat too specific to the particular gSCAN task. For instance, it considers particularly disentangling the representation of each step to direction, action and manner components. I would hope that a general approach will discover a disentangled representation that allows compositional generalization on its own, rather than being hand-engineered for the particular dataset, especially given its relative simplicity.

For the entropy regularization idea, while it may allow for compositional generalization, it may reduce the model ability to capture trends in the training data, and so it may produce too "extreme" representations that can’t account for correlations within the data, and therefore I suspect it may not work well for more complex problems beyond gSCAN.

Comments and questions
- The related work sections gives background about the subject and used dataset but has only one short like about competing approaches. It will be good to move the general text from the related work section to the introduction section and instead add a bit more detailed description about prior approaches for the problem and how they differ from the new method.
- The experiments section doesn’t provide any details about the baselines either, so overall a more detailed comparison to existing approaches or putting the paper in the context of prior work is really missing.
- Page 2: "Another related work is independent disentangled representation (Higgins et al., 2017; Locatello et al., 2019), but they do not address compositional generalization." -> Why aren’t they? The main advantage of disentangled representation is their ability to generalize to combinations of properties outside the training distribution. For instance, if you encode separate features for color and shape, you may learn to generalize to any combination of them even if the training data didn’t cover all of them, since your representation inherent compositional structure that separates out these two properties may prevent capturing spurious correlations of the two properties and encourage generalization to combinations of them.
- The description of the task is not clear enough. A more formal/mathematical definition of the task will be useful, especially if letters are presented for e.g. the input command, output sequence etc it will be easier to refer back to them later in the paper. Also further description/ a couple of examples on what the actions and the manners are will be useful for those not familiar enough with gSCAN.
- "We also assume automatic collision prevention… This makes us focus on addressing grounded compositional generalization problem." - is it fair to assume that or does it simplify the problem? Are alternative approaches assume that? How is it useful?
- Page 3: "For example, when "red" and "square" do not appear together in any training sample, a model might learn that square is not red. However, this causes errors for compositional generalization in test. To avoid such case..." - Do we want to avoid such cases? A model that will learns that there is no correlation at all between pairs of properties will not work in the real world. Rather than avoiding learning the correlations it will be useful if the model will be still able to learn them but at the same time allocate some smaller probability for the case of combinations that are less common.
- The description of entropy regularization isn’t completely cleared to me. What are x_i and y_i. Does each y_i depend on all x or only xi?
- It sounds like basically entropy regularization reduces the capacity of the representation by adding noise + l2 loss on the activations. This sounds like a quite general regularization technique but is unclear to me why this encourages compositionality in particular. Further explanation of that will be helpful.
- A small comment, since the generalization to length work well but one-shot learning isn't and is still an open problem, it may make more sense to put the length subsection first and then the one-shot learning one.

Some Typos
- What are the components in output -> in the output
- other changes with ablation study -> with an ablation study
- to understand command and the environment -> the command and
- redundant dependency on input -> on the input
- Grounded SCAN (gSCAN) dataset -> the Grounded SCAN dataset
- but agent needs -> but the agent need or but agents need
- of agent and the environment -> of the agent
- to change position -> to change its position
- on addressing grounded compositional generalization problem -> addressing the problem
- Input contains command -> the input
- Output contains -> the output
- Information of color -> of the color
- design entropy regularization layer -> a/the layer
- finds correct object -> find the correct object

---

> ### Author Response · Authors · 2020-11-20
> **Reply to Review 2**
>
> Thank you for your questions and comments.
>
> Q1: It doesn't discuss almost at all existing works and mention the only very briefly, making it harder to judge the strength of the new approach as there's not enough context.
>
> A1: The approaches are quite different. So we connect them by comparing how to encode prior knowledge. The goal of this paper is to enable accurate compositional generalization in this grounded instruction learning problem, serving for analyzing and understanding the mechanism.
>
> Q2: In addition, the idea presented feels somewhat too specific to the particular gSCAN task.
>
> A2: The novelty of this paper is to find that the combination of environment interaction and
> entropy regularization helps the generalization. This is a general way to guide how to address different problems. The research of grounded compositional generalization is still at the stage of finding fundamental mechanisms in simplified cases.
>
> Q3: For the entropy regularization idea, while it may allow for compositional generalization, it may reduce the model ability to capture trends in the training data, and so it may produce too "extreme" representations that can’t account for correlations within the data, and therefore I suspect it may not work well for more complex problems beyond gSCAN.
>
> A3: For this concern, the argument of entropy regularization should be similar to that of L1 regularization which also reduces ability to capture all information, but it is widely used.
>
> Q4: The related work sections gives background about the subject and used dataset but has only one short like about competing approaches. It will be good to move the general text from the related work section to the introduction section and instead add a bit more detailed description about prior approaches for the problem and how they differ from the new method.
>
> A4: Thanks. We would look into how to arrange.
>
> Q5: The experiments section doesn’t provide any details about the baselines either, so overall a more detailed comparison to existing approaches or putting the paper in the context of prior work is really missing.
>
> A5: The approaches are quite different, and hard to compare in detail. Please also refer to A1 above.
>
> Q6: Page 2: "Another related work is independent disentangled representation, but they do not address compositional generalization." -> Why aren’t they?
>
> A6: Independent disentangled representation assumes the components are statistically marginally independent in training distribution, which means all the combinations of component (attribute) values have positive joint probability in training. So it does not fit the requirement of compositional generalization: the joint probability of some component values are zero in training.
>
> Q7: The description of the task is not clear enough.
>
> A7: The task is described in detail in the dataset paper. We would make it more clear.
>
> Q8: "We also assume automatic collision prevention…" - is it fair to assume that or does it simplify the problem? Are alternative approaches assume that? How is it useful?
>
> A8: We design that automatic collision prevention is a part of the environment interaction. Also, the main focus of the paper is not comparing with other methods, but to analyze and understand the basic mechanism for the generalization.
>
> Q9: Page 3: "For example, when "red" and "square"..." - Do we want to avoid such cases? A model that will learns that there is no correlation at all between pairs of properties will not work in the real world. Rather than avoiding learning the correlations it will be useful if the model will be still able to learn them but at the same time allocate some smaller probability for the case of combinations that are less common.
>
> A9: This is from the task design in the dataset paper, and it is a kind of compositional generalization. The dataset might be not very natural, but it was designed to study specific types of problems.
>
> Q10: The description of entropy regularization isn’t completely cleared to me. What are x_i and y_i. Does each y_i depend on all x or only xi?
>
> A10: x_i is a scalar element in input representation, and y_i is a scalar element in output representation. For ERL here, the network structure is designed so that y_i is connected only to x_i.
>
> Q11: Entropy regularization sounds like a quite general regularization technique but is unclear to me why this encourages compositionality in particular. Further explanation of that will be helpful.
>
> A11: The key of compositionality is to avoid incorrect dependency. By reducing entropy, redundant information will be removed. When the correct dependency is necessary and the incorrect dependency is not necessary (if it provides only part of information), the incorrect dependency will be removed. Please also refer to A5 for reviewer 4.
>
> Q12: Put the length subsection first and then the one-shot learning one.
>
> A12: Thanks. We like to investigate the arrangement.
>
> Thank you for pointing out the typos. We would fix them.

---

### Official Review · AnonReviewer4 · 2020-10-28
**Good results, but many questions about the system and experiments**

**Rating:** 5
**Confidence:** 3

**Review:**

Summary:

This paper proposes a new model for the gSCAN dataset (Ruis et al. 2020) which is a synthetically-generated dataset that challenges models to generalize to new compositions of attributes and objects in an instruction. The paper proposes to use "entropy regularization" as a way to enforce that spurious correlations between input tokens and output actions are not learned. The proposed model shows promising results on the gSCAN benchmark, achieving nearly 100% accuracy on all but one task.

########

Reasons for score:

While the paper achieves impressive performance on the synthetic gSCAN dataset, the paper's framing of the problem, discussion of related work, and description of the model are vague and confusing. As a reviewer, I am left with many questions and confusions about the setup and scope of this paper, and my review depends on clarifications from the authors during the discussion period.

########

Strengths:
1. The paper achieves impressive performance on the gSCAN dataset across all but one task, and describes intuitions on why the remaining task failed.
1. The paper also achieves impressive results on the target length task when compared to existing systems.
1. The proposed model is relatively simple.

########

Weaknesses:

1. My main concern about the paper is its framing of the problem and use of vague terms, especially in the introduction and abstract. See below for more suggestions as well. The paper should be more clear about the scope of compositional generalization it is targeting, especially early in the paper. Specifically, it seems to be targeting novel compositions of attributes and objects in an input instruction. However, there are other dimensions of compositional generalization which the paper does not cover. For example, generalization to new compositions of the environment itself (i.e., novel combinations of the objects, attributes, and spatial relations, as evaluated in datasets like ALFRED (Shridhar et la. 2020), Room to Room (Anderson et al. 2018)), or novel deep compositional structures in the output space (as in semantic parsing, e.g., Finegan-Dollak et al. 2018, in comparison to shallow output space compositions such as new manner/direction combinations). This paper should clarify the dimensions along which it is evaluating compositional generalization, and contrast that with other forms of compositional generalization. Some other related work which evaluates compositional generalization and are missed include VQA with Changing Priors (Agrawal et al. 2017) and CLEVR CoGenT (Johnson et al. 2017). Especially for CLEVR CoGenT, there are several systems which achieve near perfect accuracy on that compositional generalization task, such as NS-VQA (Yi et al. 2019), which are not compared against.
1. The evaluation is only performed on a single synthetic dataset. While it performs well on this benchmark, evaluation on other grounded instruction following tasks, such as some of the examples above, or even on non-instruction tasks such as image question answering, would make the results more convincing.


########

Major questions for the authors about the paper setup :

1. I am confused about where "entropy regularization" comes into play in the model. As it is written it seems to be used in two places: (1) in the actual forward-pass, the only difference is that it adds noise to an input (and the noise is zero at evaluation time), and (2) during optimization, it also minimizes the L2 norm of these layers' activations. Is this correct?
1. In the ablation setting, it's unclear to me what exactly is being ablated. If EntReg appears both in optimization as L2 regularization and in the forward pass as noise, then are these ablated independently? What is ERL replaced with when ablated? And what exactly is NodeOut -- it's not defined anywhere earlier in the paper, if it is the most critical component of the approach (as indicated by Table 2), this is concerning.

Other clarification questions for the authors:
1. What is the intuition of the EntReg component actually solving the problem of learning spurious correlations? How does it actually limit the correlations (if this is the intent) between components if it is not considering some relationship between two components?
1. What does it mean for compositional generalization to require prior knowledge on distribution change?
1. Am I correct in understanding that the command module will always result in the same output while the agent is acting in the environment? Nothing about the command module seems to change as it is not conditioned on an environment observation.
1. How does the model know the types and values of the words in the command?
1. How do "yellow squares" and "red squares" actually differ in the compositionality they require?
1. Does "relatively" test new attributes that were never seen or referred to during training? Or just new ways of referring to these attributes?
1. What's an example of "class inference"?
1. How is attention evaluated? Automatically?

########

Suggestions to clarify the presentation of the paper and the approach:

1. The related work should contrast the proposed approach or evaluation setup with the cited works. E.g., how does the proposed approach differ from the approaches used on SCAN?
1. "entropy regularization" is a more general term than it is used in this paper, so I would suggest renaming it. E.g., it is often used to prevent entropy collapse in reinforcement learning by adding an additional term in the optimization that optimizes for higher entropy output distributions.
1. I am confused about why the term "entropy" is used here, because I don't see the relationship between the function EntReg and entropy (i.e., H(x) where x is a distribution). EntReg here just seems to add some noise to an input x, which is not necessarily a distribution.
1. Several terms are vague/undefined. The term "component" is vague without a formal definition of the task being studied. Later on, I am also confused on what "a change of direction between steps" means, and why there is a direction, action, and manner. The example in Figure 1 doesn't show the output sequences or examples of different manners, and I only have an idea of what "manner" means when I read the gSCAN paper or later on in this paper that manner might include spinning, being slow/fast, etc. So, I would suggest, e.g., showing the actual possible space of outputs.
1. The proposed approach should be described more concretely in the introduction before the results of it are described.
1. Several terms used in the model description are vague and should be defined: what is a "node" x_i (Section 4.1)? What is the difference between ERL and EntReg (my reading is that ERL is an MLP with EntReg between the layers)? What is a "keyword" (as opposed to just a "word")?

Suggestions on the description of model / results:
1. The model seems to be described formally three times: in Algorithm 1, Figure 2, and Section 4.2. Its description could be more concise (e.g., by removing Algorithm 1).
1. The experiments section should contrast the proposed approach with the baselines/other systems in a meaningful way.
1. I know this is coming from the gSCAN dataset, but the names of the 8 tasks are very confusing. If it's possible to augment these with a short keyword description that accurately describes the forms of generalization they are evaluating, this would make things a lot easier to read. Particularly, upon reading the results table, the "yellow squares" and "red squares" settings aren't meaningfully different, and I have no idea what they are testing differently.
1. If Figure 3 is just a bar graph version of Table 1, it could probably be removed as it is doesn't add any information over Table 1. It's missing the numbers in the bars, the x-axis is hard to read, and it's unclear to me whether this is graphing the mean, max, etc.
1. The meaning of $k$ should be defined before Figure 3 and Table 1 are presented.

---

> ### Author Response · Authors · 2020-11-20
> **Reply to Reviewer 4 (2/2)**
>
> Q6: What does it mean for compositional generalization to require prior knowledge on distribution change?
>
> A6: Compositional generalization has multiple components, and the generalization requires recombining values of different components in a novel way. So it requires knowing what are the types of components, such as shape and color, to recombine and generalize. This is what we mean by requiring prior knowledge on distribution change for compositional generalization.
>
> It is complicated to discuss whether machines can learn the component types without component specific prior knowledge, and it is beyond the scope of this paper. However, to our best knowledge, many current algorithms have some way to use the prior knowledge. In some cases, the prior knowledge is in the design of data structure (position and color in image). Some approaches design training data distribution to make the components statistically marginally independent. Others design model architectures, regularizations and learning algorithms with the prior knowledge. So here, we discuss different ways to use the prior knowledge to explain the difference between the approaches.
>
> Q7: Am I correct in understanding that the command module will always result in the same output while the agent is acting in the environment? Nothing about the command module seems to change as it is not conditioned on an environment observation.
>
> A7: Yes. Command module depends only on the command, which does not change while the agent is acting.
>
> Q8: How does the model know the types and values of the words in the command?
>
> A8: It’s learnt automatically. We use the prior knowledge that there should be frameworks of types and values (in design of attention), but we do not have sample-wise annotations for them.
>
> Q9: How do "yellow squares" and "red squares" actually differ in the compositionality they require?
>
> A9: The required compositionality is the same. The difference is that, during training, a yellow square object is referred to as a target object, but a red square object is not. However, a yellow square object is not directly referred to by calling it “yellow square”, but it is referred to in other ways, such as “small square”.
>
> Q10: Does "relatively" test new attributes that were never seen or referred to during training? Or just new ways of referring to these attributes?
>
> A10: It does not test new attributes, and it is just new ways of referring.
>
> Q11: What's an example of "class inference"?
>
> A11: It requires compositional generalization for the combination of action type and object size (large object needs to pull / push twice to move). In training, the combination of ‘push’ and ‘square of size 3 (large)’ is held out, and they are evaluated in the test. For example,
>
> Training: “pull a red square” (the red square is of size 3). “push a red square” (the red square is of size 2)
>
> Test: “push a red square” (the red square is of size 3).
>
> Q12: How is attention evaluated? Automatically?
>
> A12: Attention is evaluated by checking whether the attention map has the highest value on the position of the ground truth target object. The ground truth target object is not used as an input for either training or inference, but it is only used for evaluating attention.
>
> Q13: I am confused about why the term "entropy" is used here, because I don't see the relationship between the function EntReg and entropy (i.e., H(x) where x is a distribution).
>
> A13: Here, the relation is not explicit. We actually do not need to know H(x), as long as the regularization reduces it (without knowing the amount). Please also refer to A5 above.
>
> Q14: EntReg here just seems to add some noise to an input x, which is not necessarily a distribution.
>
> A14: Here, a distribution (and its entropy) is defined on a dataset, not on each sample. Please refer to A5 above.
>
> Q15: Several terms used in the model description are vague and should be defined: what is a "node" x_i (Section 4.1)? What is the difference between ERL and EntReg (my reading is that ERL is an MLP with EntReg between the layers)? What is a "keyword" (as opposed to just a "word")?
>
> A15: x_i is a scalar element (node) in input representation, and y_i is an scalar element for output representation. In ERL, each scalar element is treated separately. A scalar element reduces the number of its possible values by non-linearly expanding to high dimensional space and applying entropy regularization, then mapping back to a scalar. This is designed for the case that different scalar elements correspond to different components, so that we do not want to mix them, but hope to reduce their entropy. Keywords are just words, and keywords emphasize that they are extracted by the command module.
>
> Also thank you for other suggestions. We hope to improve the paper with them.

---

> ### Author Response · Authors · 2020-11-20
> **Reply to Reviewer 4 (1/2)**
>
> Thank you for your constructive suggestions. Following are replies to the questions.
>
> Q1: My main concern about the paper is its framing of the problem and use of vague terms, especially in the introduction and abstract. The paper should be more clear about the scope of compositional generalization it is targeting, especially early in the paper.
>
> A1: Thank you. We like to investigate how to make the scope clear and precise. We would also refer to the related work.
>
> Q2: The evaluation is only performed on a single synthetic dataset. While it performs well on this benchmark, evaluation on other grounded instruction following tasks, or even on non-instruction tasks such as image question answering, would make the results more convincing.
>
> A2: This paper focuses on analyzing and understanding fundamental mechanisms with the synthetic dataset. This might be the first step towards general grounded compositional generalization tasks.
>
> Q3: I am confused about where "entropy regularization" comes into play in the model. As it is written it seems to be used in two places: (1) in the actual forward-pass, the only difference is that it adds noise to an input (and the noise is zero at evaluation time), and (2) during optimization, it also minimizes the L2 norm of these layers' activations. Is this correct?
>
> A3: Yes, it is correct. The L2 norm is computed before adding noise in this paper, but this might be not the core point.
>
> Q4: In the ablation setting, it's unclear to me what exactly is being ablated. If EntReg appears both in optimization as L2 regularization and in the forward pass as noise, then are these ablated independently? What is ERL replaced with when ablated? And what exactly is NodeOut?
>
> A4: L2 regularization and the forward pass noise are ablated together, because we see them together as one method. When ERL is ablated, the layer is just removed (it has the same input and output shape). NodeOut means using a dedicated feed-forward neural network from input to each output node in the prediction module. We then apply entropy regularization for the input of each of these networks. We designed in this way, because different output nodes may need to be computed from different input components.
>
> Q5: What is the intuition of the EntReg component actually solving the problem of learning spurious correlations? How does it actually limit the correlations (if this is the intent) between components if it is not considering some relationship between two components?
>
> A5: We first clarify about the entropy we discuss. We then describe how entropy reduction removes effective spurious correlations, and explain how EntReg reduces entropy. We hope this answers the questions.
>
> Here, we consider a (multi-dimensional) representation as a random variable, and the distribution of this random variable with all the samples in a dataset, and we discuss the entropy of this distribution. This means for one dataset, we have only one distribution and only one entropy for the distribution. Note that we are not discussing for the setting that there is a distribution for each sample, e.g. a representation is normalized (with softmax) for each sample, and it serves as a (categorical) distribution.
>
> For example, if a dataset has four samples with scalar inputs: -1, 1, 2, 2, and we have a network of y = x^2. Then we have four y values for the samples: 1, 1, 4, 4. Then, from all the y values, we get a single entropy of y as ln(2) (we do not actually compute entropy in the algorithm). In this example, there are four samples in a dataset, and we have only one value of entropy.
>
> We then discuss how entropy reduction removes effective spurious correlations. Entropy for a random variable can be roughly understood as (log of) the number of possible values. We consider a variable Y and two input components X1 and X2. Suppose Y should depend only on X1 (with full information of Y) and X2 is redundant (with partial information of Y), and both of them have two possible values. Then, if Y has effective connections to both X1, X2, Y has 4 possible values. However, if the spurious connection is not effective, Y has only 2 possible values. The reduction of entropy reduces the number of values, and since the correct connection has full information but the spurious connection only has partial information for Y, it removes the spurious connection, and keeps the correct connection.
>
> EntReg reduces entropy in the following way. The noise makes different possible values far from each other in vector space. If they are close, the noise will make them not distinguishable, so the prediction would be wrong. At the same time, norm regularization will make different possible values close to each other to reduce the region of manifold. Intuitively, these two forces squash the values, so that unnecessary values will be merged to other values. With less number of possible values, the entropy reduces.

---

### Official Review · AnonReviewer3 · 2020-10-30
**Poorly written, poorly discussed related papers, not enough novelty**

**Rating:** 4
**Confidence:** 3

**Review:**

Summary

This paper tries to address a very important problem, compositional generalization in grounded agent instruction learning. It proposes to use interactions between agent and the environment to define output components, and entropy regularization to reduce redundant dependency on input. It shows significant improvements in most of gSCAN tasks. The paper also has an ablation study that investigates the effectiveness of entropy regularization and other factors.

Strengths

Grounded compositional generalization is a key challenge to the AI community. This paper proposes entropy regularization and shows some good results compared with Seq2seq and GECA are from Ruis et al. (2020). Semantic is from Kuo et al. (2020).

Weaknesses

The paper is very hard to read. Notations is Section 4 is very confusing.


It is not clear there is any novelty in the model architecture, command, grounding and prediction modules. If not, the paper should clearly cite relevant papers.


The paper claims to be "the first work to enable accurate compositional generalization in grounded instruction learning problem, serving for analyzing and understanding the mechanism". It is not clear this is true given Ruis et al. (2020). Semantic is from Kuo et al. (2020).


The paper fails to make a connection to the two papers. It only says "we apply the prior knowledge for the interactions of agent and the environment.".


Some qualitative results in the main text or appendix would have helped to illustrate the proposed approach and offer insights on why the method is effective.


Decision

Since the paper is poorly written, not put in the proper context of the related papers, and the novelty seems to be limited to entropy regularization, my decision is rejection. I would be open to revise my decision if the authors make clear of their methods and contributions.

=====POST-REBUTTAL COMMENTS========

The authors did not make a good effort to address my comments and failed to update the paper. Therefore, I maintain my original decision.

---

> ### Author Response · Authors · 2020-11-20
> **Reply to Reviewer 3**
>
> Thank you for your feedback.
>
> Q1: The paper is very hard to read. Notations is Section 4 is very confusing.
>
> A1: We try to improve it.
>
> Q2: It is not clear there is any novelty in the model architecture, command, grounding and prediction modules. If not, the paper should clearly cite relevant papers.
>
> A2: The novelty to find that the combination of environment interaction and entropy regularization helps the generalization, as stated in the end of introduction.
>
> Q3: The paper claims to be "the first work to enable accurate compositional generalization in grounded instruction learning problem, serving for analyzing and understanding the mechanism". It is not clear this is true given Ruis et al. (2020). Semantic is from Kuo et al. (2020).
>
> A3: Here, the emphasis is on “accurate”. Many results in this paper are close to 100% accuracy, meaning it addressed the designed problems, so that it serves for analyzing and understanding the mechanism.
>
> Q4: The paper fails to make a connection to the two papers. It only says "we apply the prior knowledge for the interactions of agent and the environment.".
>
> A4: The approaches are quite different. So we connect them using the ways to encode prior knowledge.
>
> Q5: Some qualitative results in the main text or appendix would have helped to illustrate the proposed approach and offer insights on why the method is effective.
>
> A5: Thank you. We like to look into the results.

---

### Author Response · Authors · 2020-11-25
**Thank you for suggestions**

Dear reviewers,

Thank you for constructive suggestions. We are revising the paper to have significantly higher quality, and it seems we are not able meet the deadline for updating the manuscript in this conference. We hope to continue improving the work with the valuable suggestions.

Sincerely,

---

### Decision · Program_Chairs · 2021-01-07
**Final Decision**

**Decision:**

Reject

**Comment:**

This paper proposes an approach to training language instruction following agents that aims to improve their compositional generalization., by means of an entropy regularization method to reduce redundant dependency on input.

All four expert reviewers agreed that the paper is not ready for publication in its current form. Of biggest concern is the fact that the reviewers could not interpret the exposition of the method, so were unable to be sure exactly how the method worked. This can be addressed in a future submission by clearer presentation.

Another concern was that the authors only consider a single benchmark, and fail to situate the work relative to other grounded language learning tasks and datasets. Thus, reviewers were concerned about the generality of the method, and suspected it may be too specific to the gSCAN setup.

That said, the reviewers were all impressed by the strong results on the gSCAN benchmark. It strikes me that there is some interesting insight here that can be derived from this impressive performance, that may also be applicable to other grounded language learning settings. However, to make the paper acceptable for publication the authors must do a much better job of communicating how their method works, what that specific insight is and how it is relevant beyond the gSCAN dataset (ideally via direct experimentation in other settings).